# Vectorial application for the illustration of archaeological lithic artefacts using the "Stone Tools Illustrations with Vector Art" (STIVA) Method

Jacopo Niccolò Cerasoni *

Pan-African Evolution Research Group, Max Planck Institute for the Science of Human History, Jena, Germany

* cerasoni@shh.mpg.de

## Abstract

Lithic illustrations are often used in scientific publications to efficiently communicate the technological and morphological characteristics of stone tools. They offer invaluable information and insights not only on how stone raw materials were transformed into their final form, but also on the individuals that made them. Here, the "Stone Tools Illustrations with Vector Art" (STIVA) Method is presented, which involves the illustration of lithic artefacts using vectorial graphics software (Adobe Illustrator ©). This protocol follows an optimised step-by-step method, presenting ten major sections that constitute the creation of a lithic illustration: photography, vectorial software configuration, scale, outline, scar borders, ripples, cortex, symbols, composition, and export. This method has been developed to allow researchers, students and educators to create clear and competent illustrations for any application, from scientific publications to public outreach.

## Introduction

Archaeological research involves the study of past human activity, behaviour, and interaction with the environment. Although widely different approaches can be taken depending on geographic, cultural, and chronological contexts, one of the most important components of archaeological research is the study of material culture. Objects used and created in the past can offer invaluable information for the better understanding of who made them, why, and how they were used. These objects have to be represented and published to share the researchers' interpretations and hypotheses, and this is done using graphic, printable, representations.

One of the most ubiquitous types of materials retrieved in the archaeological record is stone. Due to its inorganic nature it can survive indefinitely both in open-air contexts and in below-surface deposits, and it is undoubtedly a type of raw material that has been used for the longest time in human prehistory [1]. Knapped stone objects are fundamental artefacts for the identification of human presence and behaviour, and are commonly used to identify the earliest tools produced by our ancestors. These important objects, whether for research or public engagement purposes, are therefore best represented by means of graphic representation.

**Data Availability Statement:** All relevant data are within the manuscript and its Supporting Information files.

**Funding:** This study was funded by the Pan-African Evolution Research Group, based at the Max Planck Institute for the Science of Human History, Jena, Germany.

**Competing interests:** The author declares no competing interests.

In the past decades the introduction of easy-to-use and fast imaging technologies has overtaken the world of archaeological imaging. Prior to the introduction of digital imaging technologies such as photography, photogrammetry and three-dimensional scanning, lithic illustrations were commonly drawn by artists and archaeologists. The fastest and most efficient of these modern imaging methods is certainly digital photography. Given the ease of access to digital photographic cameras and their affordability and wide-spread use, digital photographs have become the primary method for artefact depiction [2]. However, photography can be a challenging medium, as many variables must be considered when photographing lithic artefacts. Surface colour, patination, roughness, irregularities, raw material type, opacity, and background lighting are just a few of the variables that have to be reckoned with when photographing stone, and without a thorough understanding of photography it can be easy to misjudge the final product. When details and aspects of the artefacts have been obscured by improper photography, important observations and interpretations can be missed.

Photogrammetry uses digital photographs to create three-dimensional models of objects or locations. If poor quality or poorly processed digital photographs are used in photogrammetry, the problems described above are compounded and can result in unreliable digital models [3]. This method involves the modelling of three-dimensional objects that begins with the importing of two-dimensional photographs into specialised photogrammetry softwares [4]. This is followed by the processing of the photographs to create three-dimensional surfaces that are then merged into final volumetric models [5]. Furthermore, when photogrammetry is applied to small objects such as lithic artefacts, a series of issues can be encountered due to the minute nature of the topographic and textural details, often resulting in three-dimensional models lacking in high resolution.

3-D scanning can potentially suffer from different problems that are unlike those for photography and photogrammetry. 3-D scanning is a modern imaging method that includes several techniques that differ based on the specific technology used, and can highly differ in terms of cost, usability and scanned material limitations [6]. Structured light scanners are among the most commonly used 3-D scanners in archaeological contexts, and use patterns of light to identify any deformations or reflections on the surface of the scanned object. The application of structured light scanners for lithic artefacts is usually satisfactory, although it can be problematic when scanning translucent or transparent objects; these can be common features of lithic artefacts. This problem does not arise, however, when using laser scanners. These scanners create 3-D images by triangulating the positions of laser dots which are projected onto the object, in order to recreate three-dimensional surfaces. Both of these 3-D scanning techniques can be used to create high-quality lithic illustrations [7–9], both as three-dimensional objects and two-dimensional figures. Nevertheless, 3-D scanning is a costly and time-consuming method. The high cost of 3-D scanning equipment and post-processing machinery, together with the time required for the scanning process and subsequent post-processing, make this method rarely used.

In regards to lithic artefacts, the problems that might arise from inadequate photography, photogrammetry, or 3-D -scanning, can be easily resolved with a traditional lithic illustration. Such lithic illustrations offer the opportunity to select and emphasize areas or features of a lithic artefact, and there are little to no risks in obscuring what is important for developing interpretations and hypotheses. Unfortunately, traditional hand-drawn illustrations, which are still to this date the most common method for lithic illustration, can often be time consuming or expensive should an artist be hired. To resolve the issues of time and costs related to the hand illustration of lithics, the Stone Tools Illustrations with Vector Art 'STIVA' Method was developed. Although new, its application has already been tested and proved to be publishable in peer-reviewed manuscripts [10].

Taking inspiration from the traditional step-by-step processes used by archaeological illustrators [11, 12], the 'STIVA' Method transposes the depiction of lithics from pen and paper to any digital device. By using reference photographs for digital tracing, the 'STIVA' Method combines the ease of use and speed of digital photography with the representational power of hand illustrations.

The 'STIVA' protocol offers a clear step-by-step process that anyone can learn and put into practice. Nevertheless, to produce high-quality, publishable illustrations, it is important that the illustrator has a good understanding of lithic analysis. To produce objective figures, it is necessary to know what is being drawn, and to ensure that non-existent features or components are not fabricated. For this reason, it is highly advised to have a good understanding of stone tool morphology and production.

The 'STIVA' Method was developed using Adobe Illustrator ©, as it has native functions and tools that make the digital illustration of lithic artefacts easier and faster compared to other non subscription-based vectorial softwares. The 'STIVA' Method provides detailed explanations on how to navigate Adobe Illustrator ©, and offers a framework for archaeologists and others to illustrate any archaeological material.

## Materials and methods

The protocol described in this peer-reviewed article is published on protocols.io, dx.doi.org/10.17504/protocols.io.bubqnsmw and is included for printing S1 File with this article.

## Expected results

While a variety of methods for lithic illustration already exist, with the application of the 'STIVA' method it is expected that users will produce publishable and user-friendly illustrations without the dependency on hand-drawing experience and skill. With minimal practice and the access to graphic illustration softwares and hardware, anyone interested in lithics, whether for personal, educational, or professional reasons, can produce their own high-quality lithic illustrations.

Archaeological studies are at times incongruous when artefact comparisons from different sites and time periods are attempted. With the 'STIVA' protocol, one single method can be widely used when illustrating lithic artefacts from any context or chronology. This method can therefore aid with the standardisation of stone tools illustrations, offering the potential of new and invaluable comparative capabilities.

## Supporting information

**S1 File. Stone tools illustrations with vector art: The 'STIVA' method V.2 protocol.** Also available on protocols.io.
(PDF)

## Acknowledgments

I would like to thank the reviewer(s) for the invaluable comments. I also thank Emily Y. Hallett for the help with the editing of the manuscript and support on the protocol during development, and Eleanor M.L. Scerri and Huw S. Groucutt for offering me the opportunity to illustrate the lithic assemblages they studied.

## Author Contributions

**Conceptualization:** Jacopo Niccolò Cerasoni.

**Investigation:** Jacopo Niccolò Cerasoni.

**Writing – original draft:** Jacopo Niccolò Cerasoni.

**Writing – review & editing:** Jacopo Niccolò Cerasoni.

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
