## [Decision Letter · Decision Letter 0]

9 Apr 2021

PONE-D-21-05963

Vectorial application for the illustration of archaeological lithic artefacts using the “Stone Tools Illustrations with Vector Art” (STIVA) Method

PLOS ONE

Dear Dr. Cerasoni,

Thank you for submitting your manuscript to PLOS ONE. After careful consideration, we feel that it has merit but does not fully meet PLOS ONE’s publication criteria as it currently stands. Therefore, we invite you to submit a revised version of the manuscript that addresses the points raised during the review process.

All comments have to be addressed before re-submission.

We look forward to receiving your revised manuscript.

Kind regards,

Peter F. Biehl, PhD

Academic Editor

PLOS ONE

Journal Requirements:

Additional Editor Comments:

All comments have to be addressed before re-submission.

Reviewers' comments:

Reviewer's Responses to Questions

**Comments to the Author**

1. Does the manuscript report a protocol which is of utility to the research community and adds value to the published literature?

Reviewer #1: Yes

2. Has the protocol been described in sufficient detail?

Descriptions of methods and reagents contained in the step-by-step protocol should be reported in sufficient detail for another researcher to reproduce all experiments and analyses. The protocol should describe the appropriate controls, sample sizes and replication needed to ensure that the data are robust and reproducible.

Reviewer #1: Yes

3. Does the protocol describe a validated method?

Protocols should already have been demonstrated to work in the published literature. There should be at least one original research article referenced in the manuscript in which the protocol was used to generate data.

Reviewer #1: Yes

4. If the manuscript contains new data, have the authors made this data fully available?

Reviewer #1: Yes

**5. Is the article presented in an intelligible fashion and written in standard English?**

Reviewer #1: Yes

6. Review Comments to the Author

Reviewer #1: This paper presents a step-by-step guide outlining a novel method of illustrating lithic artifacts using Adobe Illustrator (STIVA Method). The guide is clear and easy to follow, allowing a wide range of archaeologists to digitally draw their own lithic artifacts. The capacity to illustrate in this manner is not new, but it is useful in that it makes people aware of the full functionality of programs such as Adobe Illustrator and makes it accessible to people who may not be familiar with the program. It also provides a framework for others to build upon to illustrate other objects found in archaeological contexts.

While the step-by-step guide is nicely put together (see minor comments below), the remainder of the manuscript needs a bit of editing and elaboration. The Introduction would be greatly improved if the author added further discussion of where the STIVA method fits in with other ways of representing lithic artifacts. For example, the author juxtaposes photography with hand illustrations, arguing that photography is often not detailed enough to capture the relevant nuances present on lithic artifacts. However, morphometrics and 3-D scanning are a growing field, and this deserves further attention. A brief discussion of the advantages and disadvantages of various methods of illustration and representation, replete with expanded references, would be welcome. The manuscript also suffers from a number of grammatical and wording issues, and I would suggest systematic copyediting to improve its flow and readability.

Specific Comments:

• Line 47: Divulgation is perhaps the wrong word. The first sentence of the introduction should be reworded.

• Line 57-58: The following is too informal: “…has been used for the longest time in human prehistory.”

• Line 102: Change “whilst” to “while”

• Can steps 22 and 23 be combined?

• Based on your discussion in step 27, the cutting of ripple marks seems to be rather subjective. This should be made clearer in step 27.1, perhaps stating why the cutting of ripple marks is needed. Any other suggestions on the best way to do this would also be welcome.

7. PLOS authors have the option to publish the peer review history of their article (what does this mean?). If published, this will include your full peer review and any attached files.

Reviewer #1: No

---

## [Author Response · Author response to Decision Letter 0]

22 Apr 2021

I would like to thank the reviewer and editor for the detailed and very welcome comments. Following the reviewer's comments, the Introduction has been expanded so to include different lithic imaging methods (i.e. photogrammetry and 3-D scanning), discussing their advantages and disadvantages. Furthermore, more references have been added in support of the several claims made towards the imaging methods described. The specific comments raised by the reviewer were all very appreciated and corrected. The manuscript was also professionally and systematically copyedited so to solve all grammatical and wording issues.

Following the editor's comments: Sections of the manuscript have been moved so to follow the PLOS ONE template. The uploaded files have been named correctly as per request. All references have been corrected and formatted appropriately.

Once more, I would like to thank the reviewer and the editor for all the constructive, positive, comments and interest for the 'STIVA' Method.

---

## [Decision Letter · Decision Letter 1]

27 Apr 2021

Vectorial application for the illustration of archaeological lithic artefacts using the “Stone Tools Illustrations with Vector Art” (STIVA) Method

PONE-D-21-05963R1

Dear Dr. Cerasoni,

We’re pleased to inform you that your manuscript has been judged scientifically suitable for publication and will be formally accepted for publication once it meets all outstanding technical requirements, including the staff editor request below my signature.

Kind regards,

Peter F. Biehl, PhD

Academic Editor

PLOS ONE

Staff Editor Comments (optional):

PLOS ONE has a requirement that published Lab Protocols link, in the Introduction section, to at least one supporting peer-reviewed publication in which the protocol was applied to generate data (see the guidelines at https://journals.plos.org/plosone/s/submission-guidelines#loc-lab-protocols). We noted that the citation below (meeting this requirement) was included in a previous version of this manuscript but it was removed in the revised version. Can you please re-include the citation, or a citation to another paper in which the protocol was used, and also comment on why the citation was removed? If you have any questions about this requirement, please email Senior Editor Joseph Donlan at jdonlan@plos.org 

Scerri EML, Niang K, Candy I, Blinkhorn J, Mills W, Cerasoni JN et al. Continuity of the  Middle Stone Age into the Holocene. Sci Rep. 2021; 11, 70. doi: https://doi.org/10.1038/s41598-020-79418-4

Reviewers' comments:

Reviewer's Responses to Questions

**Comments to the Author**

1. Does the manuscript report a protocol which is of utility to the research community and adds value to the published literature?

Reviewer #1: Yes

2. Has the protocol been described in sufficient detail?

Descriptions of methods and reagents contained in the step-by-step protocol should be reported in sufficient detail for another researcher to reproduce all experiments and analyses. The protocol should describe the appropriate controls, sample sizes and replication needed to ensure that the data are robust and reproducible.

Reviewer #1: Yes

3. Does the protocol describe a validated method?

Protocols should already have been demonstrated to work in the published literature. There should be at least one original research article referenced in the manuscript in which the protocol was used to generate data.

Reviewer #1: Yes

4. If the manuscript contains new data, have the authors made this data fully available?

Reviewer #1: Yes

**5. Is the article presented in an intelligible fashion and written in standard English?**

Reviewer #1: Yes

6. Review Comments to the Author

Reviewer #1: The article and description of the method is greatly improved, and I would recommend acceptance for publication.

7. PLOS authors have the option to publish the peer review history of their article (what does this mean?). If published, this will include your full peer review and any attached files.

Reviewer #1: No

---

## [Editor Report · Acceptance letter]

29 Apr 2021

PONE-D-21-05963R1 

Vectorial application for the illustration of archaeological lithic artefacts using the “Stone Tools Illustrations with Vector Art” (STIVA) Method 

Dear Dr. Cerasoni:

I'm pleased to inform you that your manuscript has been deemed suitable for publication in PLOS ONE. Congratulations! Your manuscript is now with our production department. 

Kind regards, 

on behalf of

Dr. Peter F. Biehl 

Academic Editor

PLOS ONE